# The Use of Ultrasound-Assisted Maceration for the Extraction of Carnosic Acid and Carnosol from Sage (*Salvia officinalis* L.) Directly into Fish Oil

**DOI:** 10.3390/molecules28166094

**Published:** 2023-08-16

**Authors:** Agnieszka M. Hrebień-Filisińska, Grzegorz Tokarczyk

**Affiliations:** Department of Fish, Plant and Gastronomy Technology, Faculty of Food Sciences and Fisheries, West Pomeranian University of Technology in Szczecin, 71-459 Szczecin, Poland; grzegorz.tokarczyk@zut.edu.pl

**Keywords:** carnosic acid, carnosol, ultrasonic maceration, active substances of sage, rosmarinic acid, green extraction method

## Abstract

The aim of the study was to examine the effect of ultrasonic maceration (U) on the extraction of carnosic acid (CA) and its derivative—carnosol (C)—directly from sage into fish oil, compared to homogenization-assisted maceration (H). It was shown that the ultrasonic maceration process (U) allowed for obtaining a macerate enriched in carnosic acid (CA) and carnosol (C), also containing rosmarinic acid (RA), total polyphenols, and plant pigments, and showing antioxidant properties (DPPH test). There was no unequivocal difference in the efficiency of extracting ingredients from sage into the oil macerate between U and H, with the use of ultrasound in most cases resulting in a greater extraction of C and less extraction of pigments from sage into the macerate than in H. The highest simultaneous contents of CA (147.5 mg/100 g) and C (42.7 mg/100 g) in the macerate were obtained after 60 min of maceration U when using a higher power (320 W). The amount of determined compounds also depended on the concentration of methanol (methanol; 70% methanol) used for the analysis. The maceration U is a simple, safe, “green method” of obtaining active substances, with a reduced number of steps, enabling an interesting application form of CA and C, e.g., for food or cosmetics.

## 1. Introduction

Carnosic acid and its derivatives are one of the strongest natural antioxidants produced by some plant species. Carnosic acid also exhibits antibacterial, antifungal, antiviral, anti-inflammatory, anticancer and neuroprotective properties [1,2,3]. It can also be a preventive and supportive measure in the treatment of many diseases, including Alzheimer’s, Parkinson’s and COVID-19 [4] as well as some cancers [2]. Recently, research has also been conducted into its anti-prion properties [5].

Carnosic acid is composed of 20 carbon atoms, 28 hydrogens and four oxygens (C_20_H_28_O_4_) (Figure 1); its structure comprises abieta-8,11,13-triene substituted by hydroxy groups at positions 11 and 12 and a carboxy group at position 20 [1]. It belongs to the class of compounds classified as secondary metabolites. It is also known as an isoprenoid and a terpene. Because carnosic acid contains a phenolic group, it is often classified as a polyphenol. However, its genesis, biosynthetic pathway, solubility properties and role are significantly different from those of typical polyphenolic compounds and rather resemble those of terpenoids such as vitamin E and carotenoids [6].

Carnosic acid has lipophilic properties [6,7] and is therefore different from most polyphenolic compounds. In addition, it is unstable, especially in organic solvents, with an acidic environment significantly increasing its stability [8]. Also, longer extraction time [9] and elevated temperature accelerate the decomposition of carnosol compounds [9,10]. However, they are much more stable in food, including fats and oils [11,12,13]. Carnosic acid is an antioxidant that scavenges peroxygen radicals, singlet oxygen and hydroxyl radicals [1]. Despite numerous studies on the antioxidant effect of carnosic acid, the exact mechanism of action is still not well known [14,15]. Probably, it can act repeatedly and “cascade” through the successive donations of hydrogen atoms, as a result of which it oxidizes and a number of its individual derivatives are formed, such as carnosol, rosmanol and isorosmanol. These derivatives have two O-phenolic hydroxyl groups in the ortho position at C_11_ and C_12_ and still show strong antioxidant properties [16]. In the mammalian organism, this acid may also act indirectly, influencing the initiation of the synthesis of endogenous “phase 2” antioxidant enzymes [3]. Its strong antioxidant properties have been noticed before. Rosemary extracts, rich in carnosic acid and carnosol, are on the list of food additives, next to synthetic, “chemical” antioxidants, and are suitable for wide use in food, cosmetics and medicines [17,18]. Rosemary, followed by sage, among all types of plants, are the richest and most important sources of carnosic acid and its derivatives. Nevertheless, these compounds have also been identified in plants of the genus *Lepechinia*, *Oreganum, Thymus* and *Hyssopus* [4,6,14,19]. Extractions with organic solvents or water are usually used to isolate active substances from plant material with relatively low yields and long extraction times, but they require the use of large volumes of solvents. Unconventional methods are also used (e.g., supercritical fluid extraction, high hydrostatic pressure extraction, pressurized liquid extraction and pulsed electric field extraction), which allow for better extraction efficiency and shorten the process time but are associated with high investment costs [20]. Recently, much attention has been paid to the protection of the natural environment and its excessive exploitation. The energy crisis in Europe observed over the last few months has clearly accelerated the green transformation and strongly motivates taking actions aimed at sustainable management of natural resources, including energy saving. Therefore, an interesting direction of research may be simple methods of obtaining active substances where the number of production stages is limited to the necessary minimum and is thus associated with low energy demand while at the same time not requiring the use of large amounts of solvents. The development of “green” technological solutions definitely meets the needs of today. This article attempts to use ultrasound to directly extract carnosic acid from sage directly into oil. This method does not use solvents, and there are no intermediate steps that would generate the amount of absorbed energy and time. In addition, the oil seems to be a suitable matrix for carnosic acid from the point of view of its stability. As a result, the obtained macerate, i.e., oil with extracted active ingredients, can be an antioxidant preparation for use, for example, in food or emulsion-based cosmetics. In previous studies, it was proven that a similar oil macerate of sage, added to fish oil, showed antioxidant properties and inhibited oxidation in it [21,22]. It was also shown that during maceration assisted by homogenization, carnosic acid and carnosol are extracted from sage into the oil phase, as their presence in the macerate identified [22]. Nevertheless, other authors obtained very good results in the extraction of active ingredients from plants into oils using ultrasound-assisted maceration [23,24,25]. Therefore, in this article, it was decided to investigate the effects of ultrasound on the extraction of active ingredients directly from sage into fish oil during maceration. The aim of the study was to investigate the effect of ultrasound, compared to maceration assisted by homogenization, on the extraction of carnosic acid as well as its main derivative, carnosol, from sage directly into fish oil.

## 2. Results

### 2.1. Characteristics of Sage

The sage used to prepare the macerates contained polyphenolic compounds and showed antioxidant activity in the DPPH test (Table 1). In the methanol (ME) and methanol–water extracts (70ME), four active compounds characteristic of this species were identified, i.e., carnosic acid (CA), carnosol (C)), rosmarinic acid (RA) and caffeic acid (COA) (Table 2 and Figure 2). Higher contents of most active ingredients were determined in methanol–water extracts (70ME) of sage than in methanolic ones; only in the case of carnosic acid (CA), its highest amounts were recorded in methanolic extracts ME (Table 2 and Figure 2).

### 2.2. Effect of Ultrasonic Maceration Compared to Homogenization-Assisted Maceration on Extraction of Active Ingredients from Sage to Oil (Content of Active Ingredients in Oil Macerates)

In order to test the level of active ingredients in oil macerates (macerate—oil with extracted sage components), in the first stage, the process of their extraction from the oil phase to solvents (methanol—ME and 70% methanol—70ME) was carried out. The active ingredients were then identified and quantified by liquid chromatography. That is, two extracts were obtained from each type of macerate: methanol (ME) and methanol–water (70ME), in which the qualitative and quantitative composition was determined. The results of chromatographic determinations of carnosic acid and carnosol in macerates are presented in Table 3, and selected chromatograms in Figure 3.

#### 2.2.1. Carnosic Acid (CA) in Macerates—Methanol Extraction (ME)

The highest carnosic acid (CA) contents were obtained in methanol (ME) extracts from macerates obtained during 5 and 60 min ultrasonic maceration (U) at a higher power of 320 W and from maceration assisted by homogenization (H), especially after 8, 11 and 13 days (Table 3). Based on the results, no unequivocal difference in the extraction of CA between maceration U and H was found. In the case of ultrasonic maceration U, the higher power (320 W) of the process resulted in a slight increase of this component in ME extracts from the macerate, compared to the extraction (U) conducted at a lower device power (200 W), in combination with a shorter time. In the second, comparative method, where maceration was assisted by homogenization (H), a slight increase in the content of CA was observed with maceration time (Table 3). Its content after 8, 11 and 13 days of maceration increased by 3.8%, 8.9% and 1.9%, respectively, compared to the starting day of maceration (day 0). As indicated by the data (Table 3) in ME extracts from macerates, significantly higher CA content was determined than in methanol–water extracts (70ME), which may indicate that ME is a better extraction medium for this compound from the oil phase than 70ME, similarly during extraction from dried sage.

#### 2.2.2. Carnosic Acid (CA) in Macerates—70% Methanol Extraction (70ME)

In methanol–water extracts (70ME), the amount of CA was significantly lower than in methanolic ones (ME), regardless of the extraction type and parameters. The structure of CA content in 70ME extracts is highly diversified. There was also no unequivocal difference in the extraction of CA from sage between both types of maceration (U and H) in 70ME extracts. In the case of maceration H, on the first day of maceration, the lowest content of CA in 70ME extracts was recorded; then, in the following days (from day 1 to day 11), this content almost doubled compared to the starting day, and on day 13 this increase was lower than in the previous days (from day 1 to day 11) (Table 3). No influence of the time and power of maceration U on the extraction of CA from sage to oil was found when analyzing 70ME extracts. CA content in these extracts from macerates obtained using ultrasound, regardless of the process parameters, was comparable to the content in macerates H after 4, 8, 11 and 13 days.

#### 2.2.3. Carnosol (C) in Macerates—Methanol Extraction (ME)

There was no significant difference in C content in ME extracts from macerates obtained in both methods (U and H)—its content ranged from 18 to 20.6 mg/100 g of macerate, regardless of the parameters and type of maceration (Table 3).

#### 2.2.4. Carnosol (C) in Macerates—70% Methanol Extraction (70ME)

The highest amount of C was determined in 70ME extracts, especially from maceration H on the initial day (time 0), as well as from macerates obtained by means of ultrasound (Table 3). Its content in the latter was almost twice as high as in macerates, where homogenization and maceration for 1 to 13 days were used. In the case of maceration H, only on the initial day, very high C content was obtained in the macerate (in 70ME extracts), from two to over four times higher than in the remaining macerates, depending on the method and process parameters. It is worth emphasizing that at the same time, in the 70ME extracts from this macerate, a significant decrease in the content of CA (almost twofold) was determined in comparison to macerates from later days (Table 3).

#### 2.2.5. Rosmarinic Acid (RA) in Macerate—70% Methanol Extraction (70ME)

During maceration from sage into oil, apart from CA and C, rosmarinic acid (RA) was also extracted, but it was determined only in 70ME extracts, while in ME extracts it was not determined (Table 4 and Figure 4). Its largest amounts were in macerates when 60 min U maceration was used at high power (320 W).

#### 2.2.6. Total Polyphenols, DPPH and Color

In macerates, the total content of polyphenols was determined using the chemical method (Figure 5), and antioxidant activity by the DPPH test (Figure 6) and color (Figure 7) were also determined. There was no unequivocal difference in the total content of polyphenols and in the antioxidant activity of DPPH between maceration U and H. The highest content of total polyphenols was obtained during maceration U, with longer extraction time (60 min) and lower power (220 W) and higher power (320 W) in both time variants (5 and 60 min). These results were comparable with H 11 and 13 days after maceration. Similar results were obtained for the antioxidant activity of DPPH.

The color in the macerates increased with the maceration time in the case of both types of macerates and with the increase of power during ultrasonic maceration U (Figure 7), while maceration U caused relatively smaller increases of this parameter than in the case of maceration H. The lowest significant color values were determined in macerates obtained using ultrasound at lower power (220 W), regardless of the length of maceration time.

## 3. Discussion

Sage (*Salvia officinalis* L.) belongs to plants rich in active ingredients with a wide range of effects, including antioxidant properties. In the leaves and upper shoots of sage used to prepare oil macerates, bioactive compounds and indicators indicating its high antioxidant potential were determined. Both methanol and methanol–water (70%) extracts of sage showed antioxidant properties and scavenged free radicals in the DPPH test, which is most likely due to the content of polyphenolic compounds. Using the liquid chromatography method, four characteristic compounds of sage were identified, i.e., carnosic acid, carnosol, rosmarinic acid and caffeic acid, where the first three compounds are among the most important compounds of sage, responsible for its bioactivity [26]. The content of carnosic acid in the tested sage (in methanol extracts) was on average 9.67 ± 0.08 mg/g of sage (10.41 mg/g DW), and carnosol (in methanol–water extracts) was 9.48 mg/g of sage (10.21 mg/g of g DW). A similar content of carnosic acid of 11.4 mg/g of sage (in tissue cultures) was obtained by Grzegorczyk et al. [27]. However, higher amounts, at the level of 14.6 mg/g DW (in *Salvia officinalis*) were obtained by Abreu et al. [28], but they received more than twenty times less carnosol (0.4 mg/g DW).

Seventy percent methanol was a more effective solvent than methanol in extracting most of the active ingredients from sage, with the exception of carnosic acid, which was found in higher amounts in sage during extraction with methanol. These results are analogous to the studies of Durling et al. [29], where during the extraction of active ingredients from sage, a higher concentration of ethanol resulted in a higher content of carnosol compounds in the extract, while a lower concentration of ethanol contributed to more efficient extraction of rosmarinic acid. Among the various compounds in sage, carnosic acid has the most lyophilic properties, while rosmarinic acid has the most polar properties [6]; hence, methanol was a better medium for extracting carnosic acid from sage tissues, and 70% methanol for rosmarinic acid, which was not extracted with pure methanol. On the other hand, carnosol, a derivative of carnosic acid, was present in a greater amount in methanol–water extracts than in methanol ones. Although carnosol, similarly to carnosic acid, has lipophilic properties, they are somewhat lower, as evidenced by the chromatograms (HPLC), which can be treated as helpful scales of lipophilicity of the components determined (Figure 2). In the liquid chromatography technique, during the separation and determination of compounds using a column with a reversed phase (hydrophobic packing), as was the case in this study, the rate at which the compounds leave the column depends on their properties. The more lipophilic the compound, the later it leaves the column; hence, its peak will be recorded later and will have a correspondingly longer retention time than a compound with polar properties. And vice versa: the more polar the compound, the faster it leaves the column; hence, its registered peak will have a correspondingly shorter retention time than a compound with lipophilic properties.

In order to prepare the macerates, the crushed sage was combined with fish oil in a mass ratio of 1 to 5.7, as this ratio enabled obtaining the most concentrated macerate with relatively the best wetting of the sage. The process of supporting the extraction of carnosic acid from sage to oil was carried out in two ways: using ultrasound U with different time (5 and 60 min) and different power of the device (200 and 320 W), and comparatively using homogenization H, where in the case of the second method after homogenization, the systems were stored for another 13 days for free maceration and tested every 2–3 days.

Maceration U and the first step of maceration H were carried out at room temperature (19–22 °C), and then after homogenization, in the case of the second step of maceration H, the next step of free maceration was carried out at 4 °C cooling conditions. Extraction at elevated temperature was intentionally not carried out, as an increase in temperature could cause a decrease in the content of carnosic acid and carnosol, which is consistent with the observations of other researchers [9]. Then, after maceration (in both methods: U and H), the oil phase was separated from sage, and carnosic acid and its derivative, carnosol, as well as rosmarinic acid, were determined in the filtered macerate. The content of total polyphenols, DPPH activity and color were also determined. For the extraction of active compounds from the oil phase, due to their various properties, methanol M and 70% methanol 70ME were used.

The use of ultrasound-assisted maceration U as well as homogenization-assisted maceration H enabled the direct extraction of carnosic acid and carnosol directly into fish oil. Rosmarinic acid was also extracted to a small extent. Carnosic acid is fat-soluble [15]; therefore, it migrated most easily from sage to oil during maceration, where it was most numerous compared to the other compounds found in the macerate. Although rosmarinic acid was most abundant in sage, its extraction into the macerate was very difficult; therefore, its small amount was determined in the macerate, which is due to the hydrophilic properties of this acid [6] and lower solubility in oils. Even lower amounts of rosmarinic acid in a similar macerate at the level of 5.57 to 6.89 mg/kg were obtained in other studies where linseed oil was macerated with basil, oregano and rosemary, but the authors used a different addition of herbs to the oil (5 g/100 g) and different type of support (shaking) [30].

In this study, caffeic acid (with hydrophilic properties) was not determined in macerates at all, which was present in sage but probably not extracted into the oil phase. In addition, the content of carnosic acid and individual active ingredients depended on the concentration of methanol used for the analysis (to leach the ingredients from the macerate). The highest content of carnosic acid was determined in methanol extracts from the macerate, and no unequivocal difference was found in its content (in these extracts) depending on the maceration method and parameters. Its highest concentration was in macerates after ultrasonic extraction U—after 5 and 60 min, at higher power (320 W), and after 8 to 11 days of maceration assisted by homogenization H.

The increase in the content of carnosic acid along with the extension of the maceration time is not relatively large; in the second week of maceration (8–13 days), depending on the day, it ranges from 1.9 to 8.9% (±1.5) compared to the initial day of maceration (time 0). In addition, taking into account various aspects, including economics and time, the best choice is to use a short 5 min ultrasonic-assisted maceration U using a higher power of 320 W and possibly homogenization-assisted maceration H without storage (maceration time 0).

The highest amount of carnosol was determined in methanol–water extracts (70% methanol) from macerates. Ultrasound-assisted maceration H has proven to be an effective method for extracting this compound from sage to oil. Its largest amounts were determined in methanol–water extracts from macerates H immediately after homogenization, but in the following days of maceration its large decrease was noted (mostly 3–4 times).

Higher amounts of carnosol in 70% methanolic extracts from the macerate than in methanolic extracts may result, on the one hand, from the fact that 70% methanol is a slightly better extraction medium than methanol for this compound, as was the case with determinations in sage (HPLC). On the other hand, they may also result from the greater instability of carnosic acid and its increased conversion to carnosol in 70% methanol than in methanol. The components of the macerates, before being determined by chromatographic methods, first required extraction from the oil phase into solvents, and until the HPLC analysis (maximum 12 h), they remained in the solvent environment. Carnosic acid is classified as an unstable compound, especially in organic solvents [12]. Analyzing the results, it was noticed that in the case of methanol–water extracts, a large decrease in carnosic acid is almost always associated with a large increase in carnosol. Therefore, it can be assumed that the increase in its amount was caused by the conversion of carnosic acid in the environment of an aqueous solution of methanol (70%) to carnosol. According to Mulinacci et al. [31], the presence of water during extraction favors the conversion of carnosic acid to the oxidized form of carnosol. However, as reported by Jacoteta-Navarro et al. [32], the decrease in carnosic acid can also increase other, smaller derivatives of carnosic acid degradation, such as epirosmanol.

It seems less likely that carnosic acid is simply not stable in oils, and its content decreasedwith maceration time. This is evidenced by previous studies [22] in which the filtered sage macerate was stored for half a year. After 2 weeks, its value decreased by only about 15% compared to the starting day, while a greater decrease of about 44% was recorded only after half a year of storage of this macerate in refrigeration conditions. Other researchers also confirm the greater stability of carnosic acid in oils than in organic solvents [12]. Therefore, in the presented experiment, extraction of carnosic acid directly into oil seems to be a very good solution; possible complications may be associated with the conditions of analysis and the selection of the appropriate medium for the determination.

Irrespective of the type of solvent used for the determination, the highest carnosic acid and carnosol contents in the macerate, respectively 147.5 mg/100 g and 42.7 mg/100 g of the macerate, were obtained after 60 min assisted maceration ultrasound U using a higher power of the device (320 W). Higher contents in the extracts after the application of ultrasounds at the level of 108.3 mg/g of carnosic acid and 49.6 mg/g of carnosol were obtained by Irakli et al. [9], where the active compounds were extracted from another species of sage directly into ethanol, not into oil. On the other hand, apart from earlier studies [22], in which during maceration assisted by homogenization, carnosic acid and carnosol were extracted from sage into the oil phase and showed antioxidant properties in fish oil, no data have been published on the extraction of carnosic acid and carnosol directly from sage to edible oils. Nevertheless, other studies have used the extraction of herbal ingredients by ultrasound directly into the oil, mainly to aromatize and enrich it with herbal ingredients and increase its oxidative stability [23,24,25,26].

In this study, sage oil macerates obtained by ultrasound U and homogenization H were also characterized by the content of total polyphenols, increased color content and antioxidant activity shown in the DPPH test. There was no unequivocal difference in the total content of polyphenols and in the antioxidant activity of DPPH between ultrasonic maceration U and assisted homogenization H. However, in the case of color, maceration U caused relatively smaller increases in this parameter than in the case of maceration assisted by H. This parameter (color) reflects the content of chlorophyll and carotenoids in the macerate, which were extracted from sage into oil. Similarly, Lu et al. [25] obtained macerates enriched with plant pigments (chlorophyll, carotenoids) and polyphenols with antioxidant properties during the aromatization of sunflower oil with herbs and spices using ultrasound-assisted maceration. 

## 4. Materials and Methods

### 4.1. Fish Oil and Sage

Fish oil (cod liver oil) (LYSI, Reykjavik, Iceland) stabilized with tocopherol was purchased at the chemist’s in Szczecin. It was stored in a refrigerator at 4 °C. Sage (*Salvia officinalis* L.) of the *Bona* variety in dry form was obtained from an herb producer in Poland. Before the study, the plant material was ground in a mill to a powdered form, with a particle diameter of up to 0.4 mm, and then stored in tight plastic bags in a dark place until analysis, but not longer than four weeks.

### 4.2. Obtaining Sage Macerates

#### 4.2.1. The Use of Ultrasound-Assisted Maceration (U)

Four versions of macerates were prepared; for this purpose, sage was mixed with fish oil in a glass bottle (sage to oil mass ratio: 1 to 5.7) and placed in an ultrasonic bath (Ultron, Dywity, Poland; frequency—40 kHz, temperature—20 °C). Different power of the device and different maceration time were used in 4 variants: time—5 min and power—200 W; time—60 min and power—200 W; time—5 min and power—320 W; time—60 min and power—320 W. Then the macerate was filtered—the oil was separated from the sage. Analyses were carried out on the filtered oil macerate.

#### 4.2.2. The Use of Homogenization-Assisted Maceration (H)

Sage was mixed with fish oil (sage to oil mass ratio: 1 to 5.7) and homogenized (1 min, rotational speed 20,000 rpm, knife homogenizer (POL-EKO, Wodzisław Śląski, Poland). The system was then placed in a dark bottle and stored at 4 °C for the purpose. Samples for analysis were taken on the day of preparation of the macerate (initial macerate, storage time 0) and after various days of maceration, the longest up to 13 days, and then filtered, similarly according to Hrebień-Filisińska and Bartkowiak [21]. Analyses were carried out on the filtered oil macerate.

### 4.3. Extraction of Active Compounds from the Macerate to Hydrophilic Phase

The macerate was dissolved in hexane and shaken with methanol and 70% methanol; then, the samples were extracted in an ultrasonic bath according to Hrebień-Filisińska and Bartkowiak [21]. Two types of extracts were obtained from each macerate: methanol (ME) and methanol–water (70ME, 70% methanol). In the obtained extracts (ME and 70ME), active ingredients were determined by chromatographic method (HPLC). In 70ME extracts, the antioxidant activity of DPPH and the total content of polyphenols were determined using a chemical method.

### 4.4. The Chemical Analyses in Macerates

The total content of polyphenols was determined according to Singleton and Rossi [33] by spectrophotometric method with the Folin-Ciocalteau reagent against gallic acid as a standard. The absorbances of the samples were measured at λ = 760 nm. The results are presented in mg of polyphenolic compounds expressed as gallic acid in 100 g of macerate.

The antioxidant activity of DPPH was determined by the method of Yen and Chen [34]. The absorbance was measured at λ = 517 nm. The calculated percent inhibition of DPPH was calculated by the formula of Rossi et al. [35]: % DPPH = 100 − (Ap/A0) × 100, where Ap—sample absorbance; A0—absorbance of the blank sample.

Color was determined according to Hrebień-Filisińska and Bartkowiak [21], using the formula: C = 1000 × (A442 + A668) where A442—absorbance of 1 cm^3^ of macerate and 10 cm^3^ of hexane at λ = 442 nm (carotenoids); A668—absorbance of 3 cm^3^ of macerate and 3 cm^3^ of hexane at λ = 668 nm (chlorophyll).

### 4.5. Physical and Chemical Analyses in Sage

The total content of polyphenols in sage was determined according to Singleton and Rossi [33] analogically as in the case of the macerate. The polyphenols were extracted from dry ground sage with 70% methanol (70M) and methanol (M) in a water bath under reflux for half an hour. After cooling, the samples were filtered and made up to 100 cm^3^ with the appropriate solvent. Two types of extracts were obtained from sage samples: M and 70M, in which polyphenols were determined by chemical method, antioxidant activity by DPPH according to Yen and Chen [34], similarly as in the case of macerates, and identification of active ingredients by chromatographic method (HPLC).

The dry weight was determined by the drying method by drying sage samples to a constant weight at 105°.

### 4.6. Identification of Activity Compounds by Liquid Chromatography (HPLC)

Identification of the active compounds (carnosic acid, carnosol, rosmarinic acid and caffeic acid) was carried out by liquid chromatography (HPLC). An Agilent 1260 Infinity II liquid chromatograph coupled with a PDA detector was used. The separation was carried out on a Nucleosil 120-5 C18 reverse phase column with dimensions 250 × 4.6 mm at ambient temperature. The mobile phase consisted of acetonitrile (solvent A) and water with 5% acetic acid (solvent B). The flow rate was kept at 0.5 mL/min. The gradient program was as follows: 15% A/85% B from 0–12 min, then changed linearly to 0% A/100% B at 30 min, followed by a change to 85% A and 15% B at 50 min, and the 85% A and 15% B system was held for another 10 min. Total analysis time was 60 min. The injection volume was 20 µL and the peaks were monitored at λ =280 and 325 nm. The macerate and sage extracts were filtered through a 0.45 Pm membrane filter prior to injection. The selection of active compounds was based on the literature [26,36]. Preparation of standards: carnosic acid, carnosol, rosmarinic, acid, caffeic acid, coumaric acid, gallic acid and apigenin were dissolved in ethanol, filtered through a 0.45 µm filter and immediately applied to the HPLC column. Only 4 compounds were identified and determined: retention times of the determined compounds [minutes]: carnosic acid: 24.9–25; carnosol—23.2–23.3; rosmarinic acid—12.1; caffeic acid—9.1.

### 4.7. Statistical Analysis

The *t*-Student test was used to assess the significance of differences between the two groups, and Tukey’s test for *p* < 0.05 was used to evaluate three or more compared groups. The results were statistically processed using the STATISTICA program.

## 5. Conclusions

Ultrasound-assisted maceration extracted the carnosic acid and carnosol from the sage directly into the oil. There was no clear difference in the extraction of active compounds between ultrasonic maceration and assisted homogenization. However, the highest contents of carnosic acid and carnosol in the macerate were obtained after 60 min of ultrasonic-assisted maceration using a higher power (320 W). A better extraction medium during the analyses for carnosic acid was methanol, and for carnosol, 70% methanol, but in this second case it may be the result of greater instability of carnosic acid in 70% methanol and better solubility of carnosol in 70% methanol than in methanol.

The use of ultrasound-assisted oil maceration is a simple, fast, safe and ecological method of obtaining carnosic acid directly from sage into oil without any intermediate stages. Oil macerates rich in carnosic acid and its derivatives can also be a good application form of these compounds due to their greater stability in oil carriers.

The presented results indicate an interesting direction of research in the development of ecological technological solutions that meet contemporary needs but still need to be expanded and improved. It would be worthwhile for future research to focus on the development of improved maceration technology in order to obtain more concentrated macerates with higher concentrations of active ingredients. This could allow for the optimization of the presented concept and obtaining improved forms of macerates suitable for a wide range of products including food, cosmetics and medicines.

## Figures and Tables

**Figure 1 molecules-28-06094-f001:**
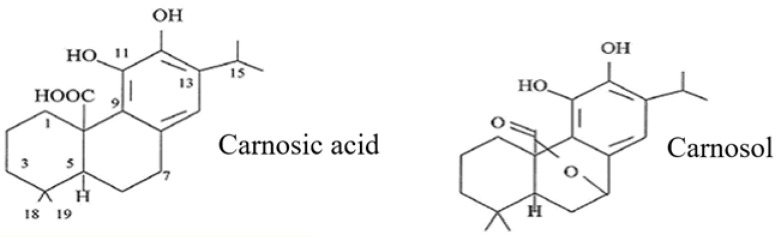
Carnosic acid and its derivative—carnosol.

**Figure 2 molecules-28-06094-f002:**
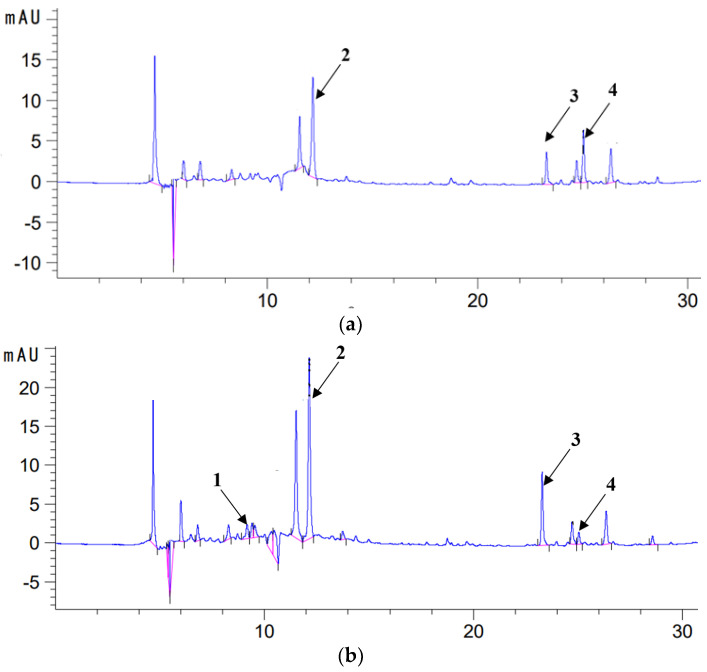
Chromatogram (HPLC) of extract with methanol (**a**) and 70% methanol (**b**) from sage (λ = 280 nm); 1—caffeic acid (COA), 2—rosmarinic acid (RA), 3—carnosol (C) and 4—carnosic acid (CA).

**Figure 3 molecules-28-06094-f003:**
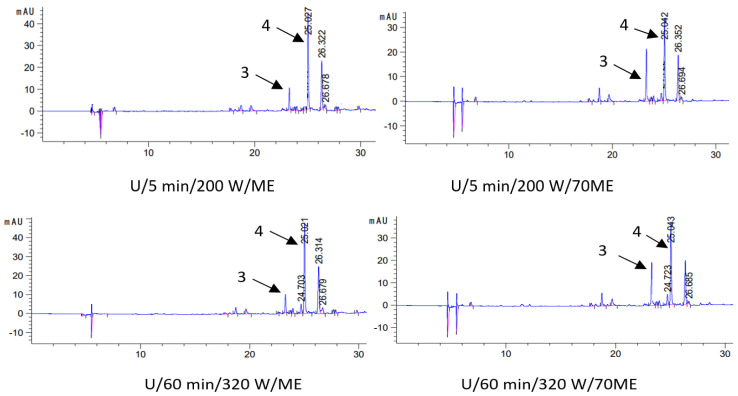
Chromatograms (HPLC) of selected macerates obtained by ultrasound-assisted maceration U (maceration times: 5 and 60 min; power: 200 and 320 W) and homogenization H (maceration times: 0 and 13 days), depending on the solvent used for the analysis (ME—methanol, 70ME—70% methanol); 3—carnosol (C) and 4—carnosic acid (CA); λ = 280 nm.

**Figure 4 molecules-28-06094-f004:**
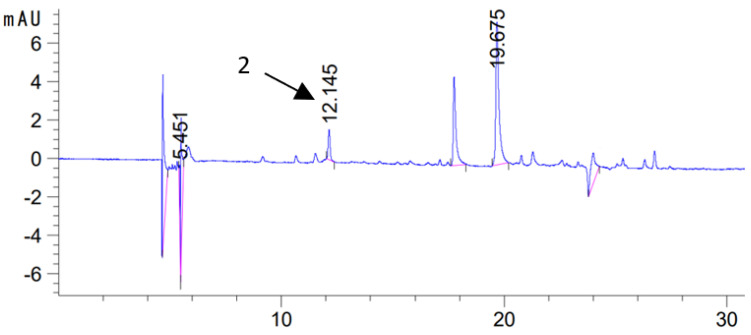
Chromatogram of sage macerate (U/60 min/320 W/70ME) obtained by 60 min ultrasonic maceration at 320 W in methanol–water extract (H/0 day/70ME)(2—rosmarinic acid, λ = 325 nm).

**Figure 5 molecules-28-06094-f005:**
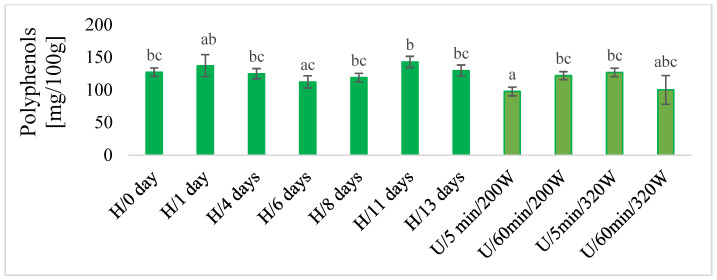
The total polyphenols in macerates obtained by ultrasound-assisted maceration U (maceration times: 5 and 60 min; power: 200 and 320 W) and homogenization H (maceration times: 0 to 13 days). ^a–c^—Different letters mean significant differences (*p* < 0.05).

**Figure 6 molecules-28-06094-f006:**
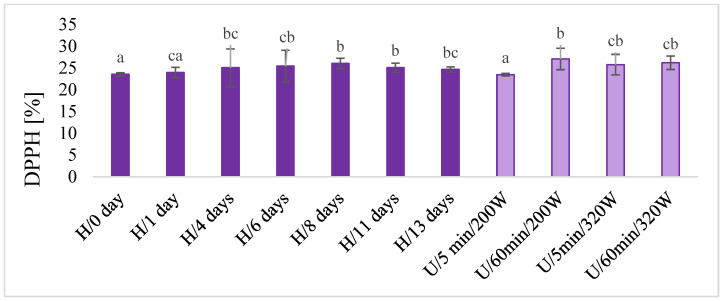
The DPPH antioxidant activity in macerates obtained by ultrasound-assisted maceration U (maceration times: 5 and 60 min; power: 200 and 320 W) and homogenization H (maceration times: 0 to 13 days). ^a–c^—Different letters mean significant differences (*p* < 0.05).

**Figure 7 molecules-28-06094-f007:**
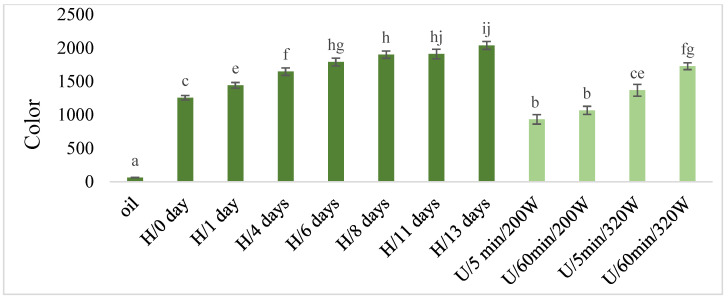
The color in macerates obtained by ultrasound-assisted maceration U (maceration times: 5 and 60 min; power: 200 and 320 W) and homogenization H (maceration times: 0 to 13 days). ^a–c, e–j^—Different letters mean significant differences (*p* < 0.05).

**Table 1 molecules-28-06094-t001:** The content of selected components of sage extracted with methanol (ME) and 70% methanol (70ME).

Dry Matter [%]	Polyphenols (mg/100 g)		DPPH [%]	
	ME	70ME	ME	70ME
92.8 ± 0.04	5746.8 ± 39.4 ^a^	7130.1 ± 157.6 ^b^	21.1 ± 0.2 ^a^	30.6 ± 0.8 ^b^

^a,b^—Values with different lowercase letters within the same parameter differ significantly (*p* < 0.05).

**Table 2 molecules-28-06094-t002:** Active compounds from sage extracted with methanol (ME) and 70% methanol (70ME) identified by liquid chromatography (HPLC).

Active Compound	Retention Time [min]	ME	70ME
Caffeic acid (COA) [mg/g]	9.1	-	1.28 ± 0.04
Rosmarinic acid (RA) [mg/g]	12.1	14.16 ± 1.32 ^a^	22.87 ± 0.02 ^b^
Carnosol (C) [mg/g]	23.3	4.42 ± 0.11 ^a^	9.48 ± 0.08 ^b^
Carnosic acid (CA) [mg/g]	25.0	9.67 ± 0.08 ^a^	1.95 ± 0.11 ^b^

^a,b^—Values in row marked with different lowercase letters differ significantly (*p* < 0.05).

**Table 3 molecules-28-06094-t003:** The content of carnosic acid (CA) and carnosol (C) in macerates obtained by ultrasound-assisted maceration U (maceration times: 5 and 60 min; power: 200 and 320 W) and homogenization H (maceration times: 0 to 13 days), depending on the solvent used for the analysis (ME—methanol, 70ME—70% methanol).

Maceration Method	Carnosic Acid [mg/100 g]		Carnosol [mg/100 g]	
	ME	70ME	ME	70ME
U/5 min/200 W	136.7 ± 4.9 ^b A^	110.6 ± 11.7 ^d c B^	18.2 ± 2.0 ^A^	34.4 ± 8.5 ^c g B^
U/60 min/200 W	137.8 ± 6.9 ^b d A^	111.5 ± 15.5 ^d c B^	20.6 ± 4.9 ^b A^	41.3 ± 12.1 ^c B^
U/5 min/320 W	145.5 ± 5.7 ^d a A^	120.7 ± 9.3 ^d c B^	19.5 ± 1.7 ^b A^	32.0 ± 7.9 ^c g B^
U/60 min/320 W	147.5 ± 11.5 ^d a A^	107.4 ± 9.4 ^d c B^	20.0 ± 1.4 ^b A^	42.7 ± 7.0 ^c B^
H/0 day	146.0 ± 2.3 ^d A^	67.6 ± 1.0 ^a B^	19.1 ± 2.3 ^A^	86.3 ± 1.0 ^a B^
H/1 day	139.5 ± 2.1 ^b A^	130.8 ± 5.5 ^b B^	20.5 ± 0.9	22.5 ± 2.2 ^b g^
H/4 days	145.8 ± 1.6 ^d A^	125.0 ± 11.7 ^b d B^	18.0 ± 1.6	23.5 ± 6.4 ^b g^
H/6 days	141.2 ± 1.0 ^b A^	135.4 ± 0.9 ^b B^	18.4 ± 0.5	19.4 ± 1.7 ^b^
H/8 days	151.7 ± 3.2 ^a A^	125.3 ± 2.4 ^b d B^	18.4 ± 3.2	19.5 ± 2.7 ^b^
H/11 days	160.2 ± 9.6 ^a A^	125.0 ± 0.8 ^b d B^	20.4 ± 2.6	19.9 ± 1.0 ^b^
H/13 days	148.9 ± 2.0 ^a A^	101.7 ± 2.2 ^c B^	18.1 ± 0.2 ^A^	25.5 ± 3.1 ^b g B^

^a–d,g,A,B^—Values within one parameter in the same column marked with a different lowercase letter differ significantly, and values in a row marked with another capital letter differ significantly (*p* < 0.05).

**Table 4 molecules-28-06094-t004:** The content of rosmarinic acid (RA) in macerates obtained by ultrasound-assisted maceration U (maceration times: 5 and 60 min; power: 200 and 320 W) and homogenization H (maceration times: 0 to 13 days), depending on the solvent used for the analysis.

Maceration Method	Rosmarinic Acid [mg/100 g] 70ME
U/5 min/200 W	5.4 ± 0.01 ^c^
U/60 min/200 W	5.0 ± 0.18 ^c e^
U/5 min/320 W	4.3 ± 0.07 ^b^
U/60 min/320 W	6.6 ± 0.10 ^a^
H/0 day	4.5 ± 0.12 ^b^
H/1 day	4.6 ± 0.04 ^b e^
H/4 days	4.8 ± 0.10 ^b e^
H/6 days	4.5 ± 0.11 ^b^
H/8 days	5.1 ± 0.01 ^c e^
H/11 days	4.5 ± 0.11 ^b^
H/13 days	4.4 ± 0.04 ^b^

^a–c, e^—Values marked with a different lowercase letter differ significantly (*p* < 0.05).

## Data Availability

The data used to support the findings of this study are available from the corresponding author upon request.

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
