# Peer review of "The Use of Ultrasound-Assisted Maceration for the Extraction of Carnosic Acid and Carnosol from Sage (Salvia officinalis L.) Directly into Fish Oil"

_molecules, 2023, doi:10.3390/molecules28166094_

Round 1

Reviewer 1 Report

The manuscript entitled: "The use of ultrasound-assisted maceration as "green extraction method" of carnosol acid and its derivative carnosol from sage (Salvia officinalis L.) into fish oil" is about the comparison of different methods of maceration extraction of carnosol acid and its derivative carnosol from sage (Salvia officinalis L.) into fish oil. In general, the manuscript is interesting and reported important data possibly use for future industrial applications or patents. The research was well designed and the methodology aligned with the objectives of the research. There are some comments to improve the quality of the manuscript before a final decision by the editor as follows:

1- Title: the title is too long and no need to highlight "green extraction method" in the title.

2- Abstract: It is good, and needs to support the results with some quantitative data in results.

3- Keywords: Choose keywords other than the main words in the title. It will improve the visibility of the manuscript.

4- Introduction: Too long, try to make it shorter and informative. For instance, Lines 96 to 106 can bring in one paragraph.

5- Results and discussion: Clear and enough dept. Check Figure 5, some letters missed on the bars.

6- Materials and Methods: Clear and detailed. Make some common methods as brief like Antioxidant Activity, ...

7- Conclusion: It is too long. Make it short and informative. Try to justify your hypothesis and bring some future research recommendations.

8- References: If possible replace those references before 2015. You can find good articles from MDPI and replace them with those outdated articles.

Author Response

(Reviewer 1)

Thank you very much for the positive review and your time. All comments I received were very helpful in improving the article. After each comment, I posted my explanation and response to the reviewer.

1- Title: the title is too long and no need to highlight "green extraction method" in the title.

Thanks for pointing out the title. After thinking about it, I came to the conclusion that it is actually too long, so I changed the title to a shorter, more appropriate one, without emphasizing that it is a green extraction method: The use of ultrasonic maceration for the extraction of carnosic acid and carnosol from sage (Salvia officinalis L.) directly into fish oil.

2- Abstract: It is good, and needs to support the results with some quantitative data in results.

According to the reviewer's suggestion, I slightly changed the summary: I added the amounts of carnosolic acid and carnosol that were maximally extracted from sage and determined in a macerate after using a 60-minute ultrasonic maceration using a power of 320W.

3- Keywords: Choose keywords other than the main words in the title. It will improve the visibility of the manuscript.

Thank you for that attention. As suggested by the reviewer, I changed the keywords.

4- Introduction: Too long, try to make it shorter and informative. For instance, Lines 96 to 106 can bring in one paragraph.

At the reviewer's suggestion, I shortened the introduction as much as possible, and wrote lines 96 to 106 in one paragraph.

5- Results and discussion: Clear and enough dept. Check Figure 5, some letters missed on the bars.

According to the reviewer's suggestion, I checked and corrected Figure 5. In fact, when marking significant differences, I missed a few letters.

6- Materials and Methods: Clear and detailed. Make some common methods as brief like Antioxidant Activity, ...

Thank you for that attention. According to the reviewer's suggestion, I shortened the methodology about the antioxidant activity of DPPH and the content of total polyphenols.

7- Conclusion: It is too long. Make it short and informative. Try to justify your hypothesis and bring some future research recommendations.

Thanks for pointing out the conclusion. As suggested, I shortened the conclusion and improved it so that it clearly justifies the hypothesis, adding recommendations for future research.

8- References: If possible replace those references before 2015. You can find good articles from MDPI and replace them with those outdated articles.

As suggested, I've replaced older references before 2015 with more recent ones, from recent years, where possible. She left only those before 2015 that were necessary and irreplaceable.

Reviewer 2 Report

The article deals with ultrasound-assisted extraction in the absence of solvent using oil directly.

The article is interesting but needs some corrections.

Here are my comments.

1.       the title needs to be corrected, it is not very clear and carnosol is used instead of carnic

2.       in some cases there are typo errors, such as "Ca" was used several times instead of "CA"

3.       in section 2.2.4 we are talking about the content of C, but it is repeatedly referred to as CA

4.       On the basis of which criterion was it chosen to recognize only carnosic acid, carnosol, rosemary acid, caffeic acid, coumaric acid, gallic acid, apigenin and not other polyphenols? Please make it explicit in the text

5.       I find it unnecessary to insert the images of the HPLC chromatograms in the text, the tables are enough. If you are really interested in inserting them, you could think of support materials

6.       why did you choose to use methanol which is considered a toxic solvent? it is not possible to understand well the purpose of this choice

I advise authors to have the article re-read by a native speaker

Author Response

(Reviewer 2)

Thank you very much for the positive review and your time. All comments I received were very helpful in improving the article. After each comment, I posted my explanation and response to the reviewer.

  1. the title needs to be corrected, it is not very clear and carnosol is used instead of carnic

Thanks for pointing out the title. I actually got the names wrong. According to the reviewer's suggestion, I changed the title to a shorter, more appropriate one and of course I used the correct names of the compounds: carnosic acid and not carnic acid

Title: The use of ultrasonic maceration for the extraction of carnosic acid and carnosol from sage (Salvia officinalis L.) directly into fish oil.

  1. in some cases there are typo errors, such as "Ca" was used several times instead of "CA"

Thanks for pointing that out, I did use the wrong symbols on a few occasions.

Of course, I corrected all these errors. I tried to do it very carefully.

  1. in section 2.2.4 we are talking about the content of C, but it is repeatedly referred to as CA

It actually looked like that, but it was the result of the wrong symbol for Ca instead of C. Of course I checked and corrected it. Now section 2.2.4 is correct. Thank you for that attention. 4

  1. On the basis of which criterion was it chosen to recognize only carnosic acid, carnosol, rosemary acid, caffeic acid, coumaric acid, gallic acid, apigenin and not other polyphenols? Please make it explicit in the text

In the research, I wanted to determine carnosolic acid and carnosol, because these compounds are mainly responsible for the antioxidant properties of sage. And it worked. In addition, I decided to check whether there are other compounds in the tested sage and extracted into the oil, since the publications indicate that they may be present in sage. The choice of standards for hplc determination of ingredients was made on the basis of previous published studies, as well as their physical availability and research funding. I selected the patterns based on the following references:

-Bianchin, M.; Pereira, D.; Almeida, J.d.F.; Moura, C.d.; Pinheiro, R.S.; Heldt, L.F.S.; Haminiuk, C.W.I.; Carpes, S.T. Antioxidant Properties of Lyophilized Rosemary and Sage Extracts and its Effect to Prevent Lipid Oxidation in Poultry Pátê. Molecules 2020, 25, 5160. -Cuvelier, M.E.; Richard, H.; Berset, C. Antioxidative activity and phenolic composition of pilot-plant and commercial extracts of sage and rosemary. J. Am. Oil Chem. Soc. 1996, 73, 645–652.

At the suggestion of the reviewer, I marked it in the text.

  1. I find it unnecessary to insert the images of the HPLC chromatograms in the text, the tables are enough. If you are really interested in inserting them, you could think of support materials

I want these chromatograms to stay. It seems to me that they are needed to better illustrate and understand the results and the phenomena taking place, the more so that in the discussion I devoted a small part to their analysis. I fear that the lack of these chromatograms may make it difficult for the reader to understand the phenomena I describe in the discussion. I hope the reviewer understands my point of view. I worked on better presentation of these chromatograms, so most of the images have been changed.

  1. why did you choose to use methanol which is considered a toxic solvent? it is not possible to understand well the purpose of this choice

Yes, of course, methanol is a toxic solvent. But my earlier research and optimization of the extraction of active ingredients was carried out on methanol, which is why I continued the above research on this solvent. However, in the near future I will consider changing methanol to a safer one, e.g. ethanol. However, it takes time, money and a lot of new research to have good results.

Comments on the Quality of English Language

I advise authors to have the article re-read by a native speaker

At the suggestion of the reviewer, the article was re-read by a native speaker, in fact there were some unfortunate wordings that were corrected in the text.
